



# Evaluation of the Lattice Boltzmann Method for wind modelling in complex terrain

Alain Schubiger[1], Sarah Barber[1], and Henrik Nordborg[1]

[1]University of Applied Sciences Rapperswil (HSR)
[1]Oberseestrasse 10, 8640 Rapperswil CH

**Correspondence:** A.Schubiger (alain.schubiger@hsr.ch)

**Abstract.** The worldwide expansion of wind energy is making the choice of potential wind farm locations more and more difficult. This results in an increased number of wind farms being located in complex terrain, which is characterised by flow separation, turbulence and high shear. Accurate modelling of these flow features is key for wind resource assessment in the planning phase, as the exact positioning of the wind turbines has a large effect on their energy production and life time. Wind

modelling for wind resource assessments is usually carried out with the linear model WAsP, unless the terrain is complex, in which case Reynolds-Averaged Navier-Stokes (RANS) solvers such as WindSim and ANSYS Fluent are usually applied. Recent research has shown the potential advantages of Large Eddy Simulations (LES) for modelling the atmospheric boundary layer and thermal effects; however, LES is far too computationally expensive to be applied outside the research environment. Another promising approach is the Lattice Boltzmann Method (LBM), a computational fluid technique based on the Boltzmann

transport equation. It is generally used to study complex phenomena such as turbulence, because it describes motion at the microscopic level in contrast to the macroscopic level of conventional Computational Fluid Dynamics (CFD) approaches, which solve the Navier-Stokes (N-S) equations. Other advantages of LBM include its efficiency, near ideal scalability on High Performance Computers (HPC) and its ability to easily automate the geometry, the mesh generation and the post-processing of the geometry. However, LBM has not yet been applied to wind modelling in complex terrain for wind energy applications,

mainly due to the lack of availability of easy-to-use tools as well as the lack of experience with this technique.. In this paper, the capabilities of LBM to model wind flow around complex terrain are investigated using the Palabos framework and data from a measurement campaign from the Bolund Hill experiment in Denmark. Detached Eddy Simulations (DES) and LES in ANSYS Fluent are used as a numerical comparison. The results show that there is in general a good agreement between simulation and experimental data, and LBM performs better than RANS and DES. Some deviations can be observed near the

ground, close to the top of cliff and on the lee side of the hill. The computational costs of the three techniques are compared and it has been shown that LBM can perform up to 5 times faster than DES, even though the set-up was not optimised in this initial study. It can be summarised that LBM has a very high potential for modelling wind flow over complex terrain accurately and at relatively low costs, compared to solving the N-S conventionally. Further studies on other sites are ongoing.



## 1   Introduction

In order to assess the wind resource for both the planning and the assessment of wind farms, measurements and simulations of the prevailing wind conditions are required. Simulations are especially crucial in the observation of flows over complex terrain due to the large impact of steep inclines on the flow conditions. If the terrain shows only weak topographic changes

or low hills, linear models can be used to make fast and sufficiently accurate yield forecasts (Berg and Kelly, 2019). The extremely low computational effort and ease of use makes such models the current industry standard. Due to their simplified formulation, however, such models fail in complex terrain and the predictions can be unreliable (Bowen and Mortensen, 1996). For complex flows, non-linear methods that solve the Navier-Stokes equations are better suited. The successful use of Reynolds-Averaged Navier-Stokes (RANS) models has been demonstrated in several studies (e.g. Ferreira et al. (1995), Maurizi et al.

(1998), Kim et al. (2000), Castro et al. (2003)), and they are being used increasingly in the industry. This is reflected by the recent development of wind energy specific tools using RANS based Computational Fluid Dynamics (CFD), including WAsP-CFD (Bechmann, 2012) and WindSim (Dhunny et al., 2016). The RANS equations govern the transport of the averaged flow quantities, with the whole range of the scales of turbulence being modelled. The RANS based modelling approach therefore greatly reduces the required computational effort and resources, and is widely adopted for practical engineering applications.

A more detailed modelling of turbulence is possible using Large Eddy Simulations (LES). LES lies between Direct Numerical Simulations (DNS) and turbulence closure schemes. The idea of this method is to compute the mean flow and the large vortices exactly. The small-scale structures are not simulated, but their influence on the rest of the flow field is parameterised by a heuristic model. However, the computational effort and the demands on the computational mesh increase drastically compared to RANS simulations, due to the need to resolve the small and important dynamic eddies in the boundary layer. Recent studies

of the Bolund Hill blind test also show that it is still a great challenge to achieve sufficiently accurate predictions using LES simulations (Bechmann et al. (2011), Diebold et al. (2013)). This is because to accurately resolve the small-scale turbulent structures near walls at high Reynolds numbers, a extremely fine grid resolution is required.

The Detached Eddy Simulation (DES) method is a combination of LES and RANS. With this method, the flow is mostly calculated by LES, but the flow and vortices in wall regions are modelled by RANS. This method promises a strong reduction

of the computational effort and the mesh requirements compared to LES. In addition, boundary layer modelling using RANS models makes it possible to use surface roughness models (Bechmann and Sørensen, 2010).

An alternative to solving the N-S equations with great potential is the Lattice Boltzmann Method (LBM). LBM has become more and more popular in recent years and is being continuously developed further. LBM has also been used successfully for initial studies in the field of wind energy. Most of these studies focus on the simulation of flows around wind turbines

and wind farms or analyse the wake behaviour of turbines (e.g. Deiterding and Wood (2016), Asmuth et al. (2019)). Studies have shown that LBM is a valid alternative to conventional CFD methods and has many advantages. The main advantage of the method is its almost ideal scalability. This makes the application on High Performance Computers (HPC) attractive, but Graphics Processing Unit (GPU) based LBM codes have also been implemented recently (Schönherr et al. (2011), Onodera and Idomura (2018)). This makes it possible to perform computationally intensive LES simulations on a simple desktop in a





reasonable time (Asmuth et al., 2019). However, LBM has not yet been assessed for the calculation of wind fields in complex terrain for wind energy applications.

The goal of this present paper is therefore to evaluate the capabilities of LBM for wind modelling in complex terrain. ANSYS Fluent is used as reference for comparisons, using both a RANS and a DES approach. The paper starts with a brief introduction

of the theories behind LBM and the conventional Navier-Stokes based CFD calculations in Section 2, then introduces the simulation method applied in Section 3, discusses the results in Section 4, and finishes with the conclusions in Section 5.

## 2   Lattice Boltzmann Method theory

### 2.1   Numerical method and governing equations

Interest in LBM has been growing in the past decades as an efficient method for computing various fluid flows, ranging from

low-Reynolds-number flows to highly turbulent flows (e.g. Chen and Doolen (1998), Filippova et al. (2001)). The first LBM models struggled with high-Reynolds-number flows due to numerical instabilities. To solve this problem, various adaptions such as regularised Finite Difference (Latt and Chopard, 2006), multiple relaxation time (MRT) (d'Humieres, 2002) or entropic methods (Ansumali and Karlin, 2000) have been developed.

LBM has the following advantages over NS: 1. A linear equation with only local instability, making it more stable and

perfectly scalable, 2. The dissipation is introduced locally by the collision term and does not depend on the lattice, and 3. the relaxation time includes both the regular viscous effects and its higher order modifications. A description of LBM can be found, for example in Chen and Doolen (1998). The governing equations describe the evolution of the probability density of finding a set of particles with a given microscopic velocity at a given location:

$$f_i(\boldsymbol{x} + \boldsymbol{c_i}\Delta t, t + \Delta t) = f_i(\boldsymbol{x},t) + \Omega_i(\boldsymbol{x},t) \tag{1}$$

for $0 \le i < q$, where $\boldsymbol{c_i}$ represents a discrete set of $q$ velocities, $f_i(\boldsymbol{x},t)$ is the discrete single particle distribution function corresponding to $c_i$ and $\Omega_i$ an operator representing the internal collisions of pairs of particles. Macroscopic values such as density $\rho$ and the flow velocity $u$ can be deduced from the set of probability density functions $f_i(\boldsymbol{x},t)$, such as:

$$\rho = \sum_{i=0}^{q-1} f_i, \qquad \rho\boldsymbol{u} = \sum_{i=0}^{q-1} f_i\boldsymbol{c_i} \tag{2}$$

The set of allowed velocites in LBM is restricted by conservation of mass and momentum and by rotational symmetry

(isotropy). Some of the most popular choices for the set of velocities are D2Q9 and D3Q27 lattices, referring to nine velocities in 2D and 27 velocities in 3D. For both of these lattices, the speed of sound in lattice units is given by $cs = 1/\sqrt{3}$ (Succi, 2001). The collision operator $\Omega_i$ is typically modelled with the Bhatnagar–Gross–Krook (BGK) approximation (Bhatnagar et al., 1954). It assumes that the fluid locally relaxes to equilibrium over a characteristic timescale $\tau$. The relaxation time $\tau$ determines how fast the fluid approaches equilibrium and is thus directly dependent on the viscosity of the fluid. The





corresponding equilibrium probability density function $f_i^{(eq)}$, is defined as:

$$\Omega_i = -\frac{1}{\tau}\left[f_i(\boldsymbol{x}, t - f_i^{(eq)}(\boldsymbol{x}, t))\right] \tag{3}$$

The equilibrium distribution function $f_i^{(eq)}$ is a local function that only depends on density and velocity in the isothermal case. It can be computed thanks to a second order development of the Maxwell–Boltzmann equilibrium function (Qian, 1992):

$$f_i^{(eq)} = w_i \rho\left[1 + \frac{\boldsymbol{c_i} \cdot \boldsymbol{u}}{c_s^2} + (\frac{\boldsymbol{c_i} \cdot \boldsymbol{u}}{2c_s^2})^2 - \frac{\boldsymbol{u}^2}{2c_s^2}\right] \tag{4}$$

where $w_i$ refers to the gaussian weights of the lattice. A Chapman–Enskog expansion, based on the assumption that $f_i$ is given by the sum of the equilibrium distribution plus a small perturbation $f_i^1$:

$$f_i = f_i^{(eq)} + f_i^{(1)}, with f_i^{(1)} \ll f_i^{(eq)} \tag{5}$$

can be applied to equation 1 in order to recover the exact N-S equation for quasi-incompressible flows in the limit of long-wavelength (Chapman et al., 1990). The pressure is thus given by $p = c_s^2 \rho$ and the kinematic viscosity is linked to the BGK relaxation parameter through:

$$\nu = c_s^2\left(\tau - \frac{1}{2}\right) \tag{6}$$

The numerical scheme is divided in two steps:

– A collision step where the BGK model is applied:

$$f_i(\boldsymbol{x}, t + \frac{1}{2}) = f_i(\boldsymbol{x}, t) + \frac{1}{\tau}\left[f_i^{(eq)}(\boldsymbol{x}, t) - f_i(\boldsymbol{x}, t)\right] \tag{7}$$

– A streaming step:

$$f_i(\boldsymbol{x} + \boldsymbol{c_i}, t + 1) = f_i(\boldsymbol{x}, t + \frac{1}{2}). \tag{8}$$

In the collision step particle populations interact and change their velocity directions according to scattering rules. This operation is completely local which makes LBM well suited for parallelism. The streaming step consists of an advection of each discrete population to the neighbour node located in the direction of the corresponding discrete velocity. Since a boundary node has less neighbours than an internal node, some populations are missing at the boundary after each iteration. These populations need to be reconstructed, which is the purpose of the implementation of boundary conditions in LBM.

## 2.2 Turbulence modelling

Turbulence leads to the appearance of eddies with a wide range of length and time scales, which interact with each other in a dynamically complex way. Given the importance of the avoidance or promotion of turbulence in engineering applications, it is no surprise that a substantial amount of research effort is dedicated to the development of numerical methods to capture the important effects due to turbulence. The methods can be grouped into the following four categories:





- Turbulence models for Reynolds-Averaged Navier–Stokes (RANS) equations

- Large Eddy Simulation (LES)

- Dettached Eddy Simulation (DES)

- Direct Numerical Simulation (DNS)

In this work LES was applied for the LBM simulations. LES is an intermediate form of turbulence calculation which simulates the behaviour of the larger eddies. The method involves spacial filtering, which passes the larger eddies and rejects the smaller eddies. The effects on the resolved flow (mean flow plus large eddies) due to the smallest, unresolved eddies are included by means of a so-called sub-grid scale model. It is assumed that the sub-grid scales have the effect of a viscosity correction, which is proportional to the norm of the strain-rate tensor at the level of the filtered scales, $\nu = \nu_0 + \nu_T$. $\nu_T$ is defined as:

$$\nu_T = C^2 |S| \tag{9}$$

where $C$ is the Smagorinsky constant and the tensor-norm of the strain rate is defined as $|S| = \sqrt{S:S}$. The value of the Smagorinsky constant depends on the physics of the problem and usually varies between 0.1 and 0.2 far from boundaries (Davidson, 2015).

## 3  Simulations

### 3.1  The Bolund Hill Experiment

The Bolund field campaign took place from December 2008 to February 2009 on the Bolund Hill in Denmark. Bolund Hill is a 130 m long (east–west axis), 75 m wide (north–south axis) and 11.7 m high hill, situated near the Risø Campus of the Technical University of Denmark. Details of the experiment are described in Bechmann et al. (2011). The campaign showed dominant wind directions from the west and south-west. Thus the wind has an extensive upwind fetch over the sea before encountering land, leading to a "steady" flow on the windward side of the hill. The wind first encounters a 10 m vertical cliff, after which air flows back down to sea level on the east side of the hill. A recirculation zone and a flow separation are expected due to this abrupt change of slope. During the campaign, 35 anemometers were deployed over the hill. The location of the measurement devices can be seen on Figure 1. Instrumentation included 23 sonic anemometers, 12 cup anemometers and two lidars. At each measurement location, the three components of the wind velocity vector and their variances were recorded for four different dominant wind directions, three westerly winds originating from the sea (268°, 254° and 242°) and one easterly wind originating from the land (95°). The mean wind speed during the measurements was around 10 ms$^{-1}$, leading to a Reynolds number of $Re = uh/\nu \approx 10^7$ with the free stream velocity $u = 10$ ms$^{-1}$, the hill height $h$ and the kinematic viscosity $\nu$. The measured values are ten minute averages of measurements sampled at 20 Hz for sonic anemometers.

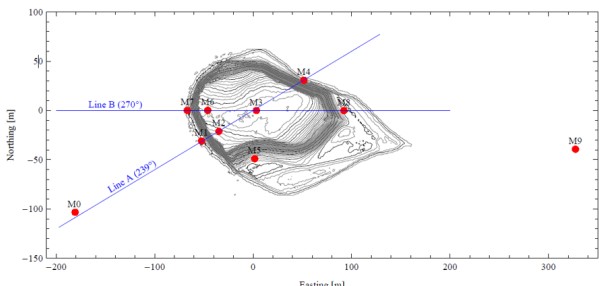

**Figure 1.** A contour map of Bolund Hill with meteorological masts denoted from M0 to M9.

## 3.2 Simulations Set-Up

### 3.2.1 Boundary Conditions

**Palabos**

The LBM flow solver used in this work was the Palabos open-source library (Latt et al., 2009). The Palabos library is a frame-
work for general-purpose CFD with a kernel based on LBM. The use of C++ code makes it easy for experienced programmers
to install and run on any machine. It is thus possible for experienced modellers to set up fluid flow simulations with relative ease
and to extend the open-source library with new methods and models, which is of paramount importance for the implementation
of new boundary conditions.

To calculate the wind fields with Palabos in this work a 525 m long (east-west axis), 250 m wide (north-south axis) and
40 m high domain with a uniform grid resolution of $\Delta x = \Delta y = \Delta z = 0.5$ m was used, leading to an total cell count of 46
million. There are no turbulence closure models or surface roughness models implemented in the Palabos library yet, therefore
the water surfaces were prescribed as free-slip bounce back nodes and the ground surfaces were modelled using Regularised
Bounce Back nodes (Malaspinas et al. (2011), Izham et al. (2011)). The bounce-back scheme in this first study was chosen
due to its simple implementation and robustness. There are more sophisticated models, like the Immersed Boundary Method
(IBM), which may provide better accuracy than the staircase approximation of bounce-back nodes, which will be investigated
in further studies.

The inlet profile was described according to the Bolund Hill Blind test specification for the westerly wind case. The logarithmic
velocity profile is defined as:

$$u(z_{agl}) = \frac{u_{*0}}{\kappa} ln(\frac{z_{agl}}{z_0}) \tag{10}$$

with $\kappa = 0.4$, the friction velocity $u_{*0} = 0.4$, the elevation above ground level $z_{agl} = z - gl$ ($gl = 0.75$ m) and the roughness
length $z_0 = 0.0003$ m . Additionally, a time varying fluctuation of the wind speed, corresponding to the given turbulence inten-
sity value, was superposed. The logarithmic wind profile was updated every second during the simulation. The Atmospheric





Boundary Layer (ABL) was considered neutral and thermal effects are therefore neglected. Each simulation was run for 600 s with a time step $\Delta t = 2.89$ ms, leading to around 10 advections times.

**Fluent**

ANSYS Fluent contains the broad, physical modelling capabilities needed to model flow, turbulence, heat transfer and reactions
for industrial applications, ranging from air flow over an aircraft wing to combustion in a furnace, from bubble columns to oil platforms, from blood flow to semiconductor manufacturing and from clean room design to wastewater treatment plants. For the Fluent simulations in this work the domain was extended to 830 m x 450 m x 60 m and two mesh refinement zones near the hill were implemented. The mesh resolution ranged from 0.5 m near the hill up to 15 m in the far-field, resulting in a total cell count of 10 million. A roughness length of $z_0 = 0.3$ mm was prescribed for the water surface and a roughness length of
$z_0 = 15$ mm for the ground surfaces. The RANS simulation was used to initialise the flow and turbulence quantities for the DES simulation. Each simulation was run for 600 s with a time step $\Delta t$ of 50 ms, leading to around seven advection times for the DES Fluent simulations. The inlet velocity was described as discussed before. The turbulent kinetic energy (TKE) at the inlet was set to $0.928 \ \mathrm{m^2 s^{-2}}$. For the DES model the Synthetic Turbulence Generator scheme was used to generate a time-dependent inlet condition. It uses a Fourier based synthetic turbulence generator. This method is inexpensive in terms of
computational time compared with the other existing methods while achieving high quality turbulence fluctuations (ANSYS, 2019).

## 4    Results and Discussion

### 4.1    Flow comparisons

The calculated velocity magnitude fields at a vertical plane through the position of met mast M3 for each measurement tech-
nique are shown in Fig. 2 and Fig. 3. Fig. 3a shows the averaged velocity magnitude over the simulation time for the RANS simulations and Fig. 3b shows the instantaneous velocity magnitude at time t = 600 s for DES. The LBM results are shown in Fig. 2, in terms of the averaged velocity magnitude over the simulation time (a) and the instantaneous velocity magnitude at time t = 600 s (b). It is interesting to note the separation region as the wind flows over the sharp edge of the hill, as well as the highly separated flow at its rear side.

### 4.2    Performance comparisons

For a quantitive comparison, the same methology is used as described by Bechmann et al. (2011) for the wind flow along the 270° axis (Case 1) as shown in Fig. 1. This involved investigating the difference between measurements and simulations after the mast M0 by comparing and quantifying the changes in the wind field as both changes in speed (so-called "speed-up") and in direction (so-called "turning"). Speed-up is defined as:

$$\Delta S_m = \frac{\langle \bar{s}/u_{*0} \rangle_{z_{agl}} - \langle \bar{s_0}/u_{*0} \rangle_{z_{agl}}}{\langle \bar{s_0}/u_{*0} \rangle_{z_{agl}}} \qquad (11)$$





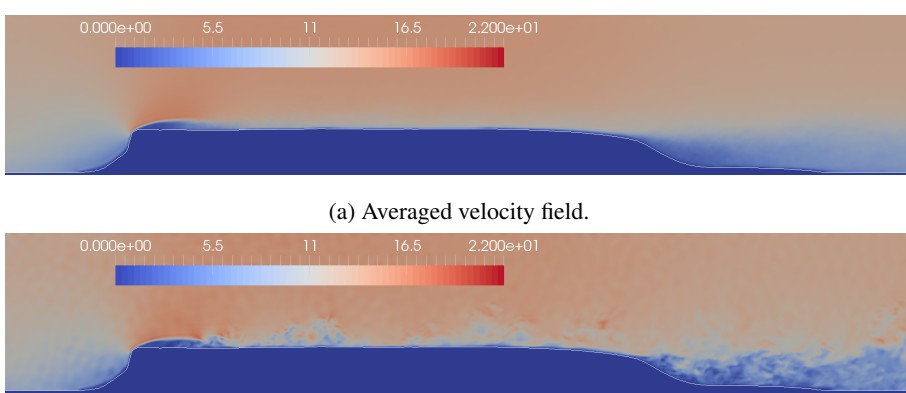

(a) Averaged velocity field.

(b) Instantaneous velocity field at t = 600 s (LBM).

**Figure 2.** Velocity field over the hill along the B line (LBM)

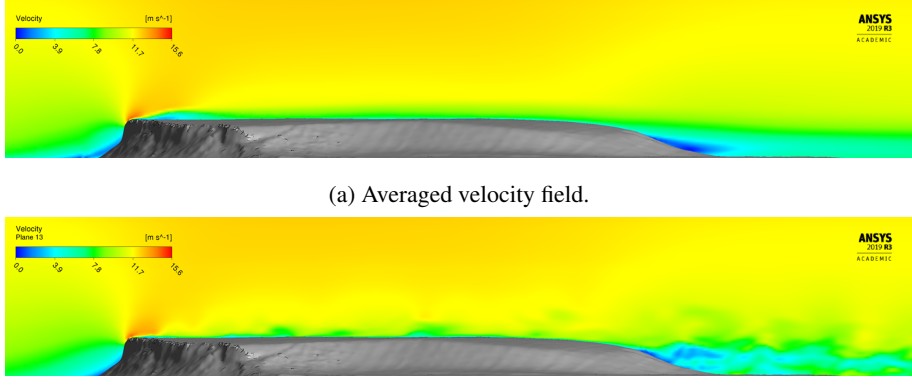

(a) Averaged velocity field.

(b) Instantaneous velocity field.

**Figure 3.** Velocity field over the hill along the B line (Fluent results)

where $\bar{s}$ is the mean wind speed at the sensor location and $\bar{s}_0$ is the mean wind speed at the mast M0. Turning is defined as the difference between the wind direction at the measurement point and that at M0. The comparison is made for two different elevations, 2 m and 5 m above the ground level and for the four masts along the B line (M7, M6, M3 and M8). The simulation results for the speed-up (Fig. 4) show good agreement with experimental data for all simulation techniques at 5 m above ground

5 level (agl), with all deviations lower than 7.1% and the average speed-up error for each simulation technique shown in Table 1. The average speed-up error is defined as:

$$R_s = 100(\Delta S_s - \Delta S_m) \tag{12}$$

where $S_m$ is the measured speed-up and $S_s$ is the simulated speed-up defined by Eq. 11. Table 1 also allows the three simulation techniques to be compared to each other. The results 2 m agl show higher deviations in general, with the average speed-up

10 error for each simulation technique shown in Table 1. The highest discrepancy can be seen at M6, which is probably due to the





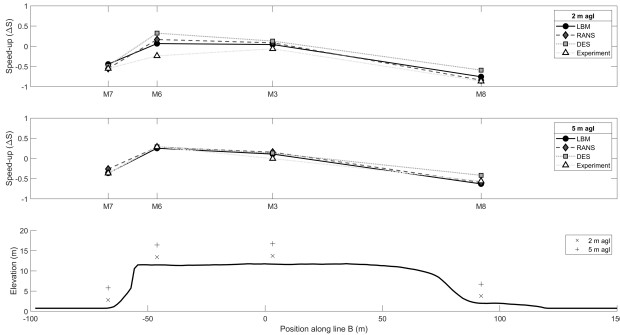

**Figure 4.** Speed-up along the Bolund Hill. Wind direction is from 270°

**Table 1.** Average Speed-up error

|  | Error at 2 m | Error at 5 m | Average error |
|---|---|---|---|
| Palabos LES | 15.7 | 0.3 | 8.0 |
| Fluent RANS | 14.6 | 5.5 | 10.0 |
| Fluent DES | 27.4 | 7.1 | 17.3 |

separation bubble observed in the velocity fields in Fig. 2a. The experiment showed reduction in wind speed at M6, whereas the simulations all show an increase in wind speed. This leads to the conclusion that the actual separation bubble is larger than the simulated one. This could be due to an error in the CAD capture of the overhang of the hill noted in previous studies (Lange et al., 2017). Furthermore, all the simulation techniques under-predicted the negative speed-up in the highly separated region

5   of M8 compared to the experiment. The reason for this is probably due to the well-known difficulty of correctly simulating the separation point in CFD. As this effect is particularly pronounced at a height of 2 m above ground, it may be due to the fact that the lower measuring points lie within the boundary layer and the used models were not able to capture the near-wall flow entirely correctly, perhaps due to the assumptions regarding surface roughness.

The most accurate overall prediction was the LBM simulation, with an averaged error of 8.0%. The RANS and DES mean

10   errors are 10.0% and 17.3%, respectively. All three methods showed more accurate results at 5 m than at 2 m above ground, as shown in Table 1.

For the turning of the wind, a similar behaviour can be observed. The results match the experimental data very well at 5 m agl, with all deviations lower than 3.0% and the average turning error for each simulation technique shown in Table 2. As for the speed-up, the deviations in turning are higher at 2 m agl, with the average turning error for each simulation technique

15   shown in Table 2. The highest discrepancy can be seen at M8. Met mast M8 is located at the lee side in the recirculation zone of the hill. All the simulation results struggle to capture the flow accurately in terms of the turning. This could be due to the inaccuracy in predicting the exact separation location on the rear of the hill, as mentioned above.   Further analysis using the





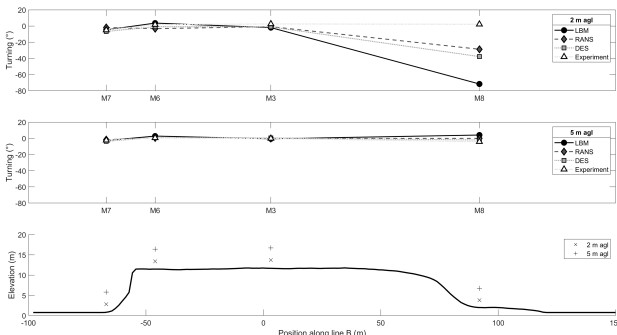

**Figure 5.** Turning along the Bolund Hill. Wind direction is from 270°

**Table 2.** Average Turning error

|  | Error at 2 m | Error at 5 m | Average error |
|---|---|---|---|
| Palabos LES | -6.2 | 0.9 | -2.7 |
| Fluent RANS | 3.0 | 0.4 | 0.2 |
| Fluent DES | -2.7 | 1.7 | -2.0 |

entire set of measurement data is shown in Fig. 6, in which a comparison between the simulation and experimental data for all three simulation methods is shown. Overall there is a good agreement between the measurements and simulated results. M2 and M6, both right after the edge of the cliff, show the biggest mismatch due to the detached flow after the edge of the hill, as discussed above. The next two figures show the ratio of simulated wind speeds to measured wind speeds as function of elevation (Fig. 7) and measurement location (Fig. 8)). The biggest deviation between the data can again be seen at lower heights and at mast M2, M6 and M8. Between the simulation methods, LBM shows the highest averaged deviation of the ratios. The DES and RANS model perform both better in this comparison. This may be due to both these models use the $SST\ k-\omega$ turbulence model and incorporate the surface roughness to calculate the near wall turbulence. The reason for the DES model performing worse than the RANS model is unclear at this point and requires further investigation.

### 4.3 Performance comparisons

### 4.4 Cost comparisons

In this section, the performance of the simulation techniques is compared in terms of the computational costs. This has been done because the overall cost of a simulation is an important factor for modellers, who need to choose the most suitable model for a given wind energy project. The results of this work have been used in order to develop a new method for helping wind modellers choose the most cost-effective model for a given project. This was done by firstly defining various parameters for



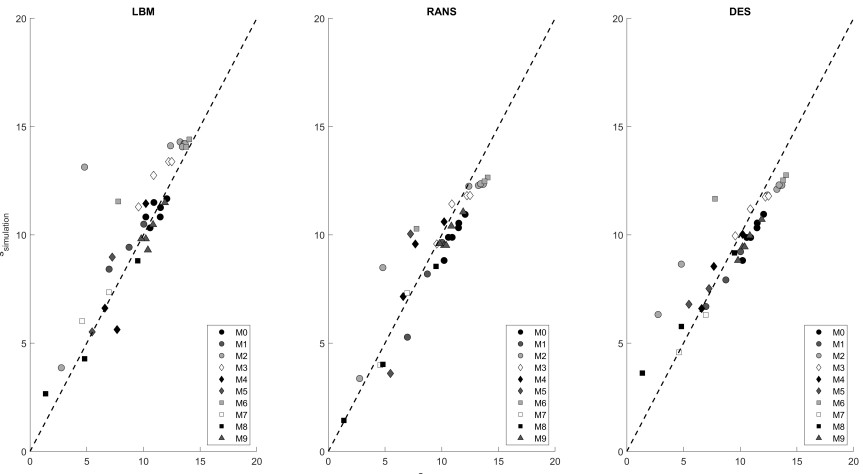

**Figure 6.** Scatter plot of wind speeds, measurement against simulation results

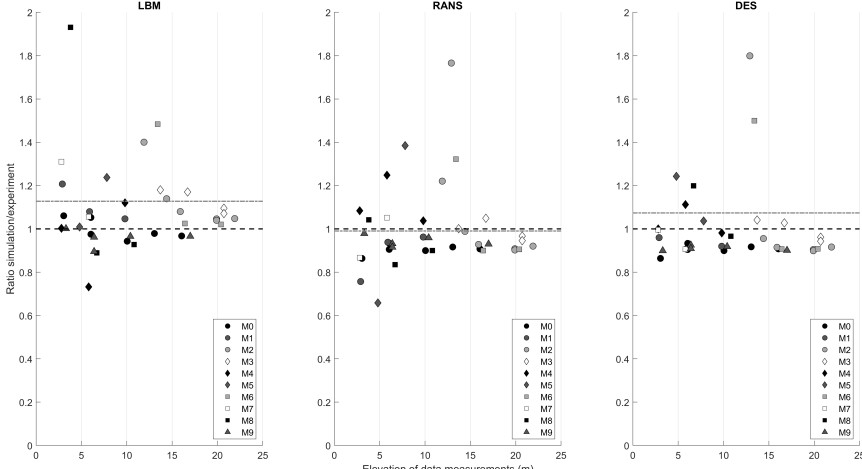

**Figure 7.** Ratio of simulation results to experimental wind speeds as function of elevation. The dotted grey line represents the average value.

predicting the skill and cost scores before carrying out the simulations as well as for calculating skill and cost scores after carrying out the simulations. Weightings were then defined for these parameters, and values assigned to them for a range of tools, including the ones applied in the present work, using a template containing pre-defined limits in a blind test. This allowed a graph of predicted skill score against cost score to be produced, enabling modellers to choose the most cost-effective model

5   without having to carry out the simulations beforehand. More details can be found in Barber et al.(in Review).

Figure 9 and Table 3 summarise the computational costs for the three different techniques applied in this paper. It can clearly be seen that the LBM performed five times faster then the DES simulation and only slightly slower than the steady


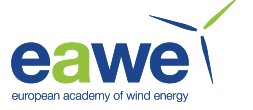


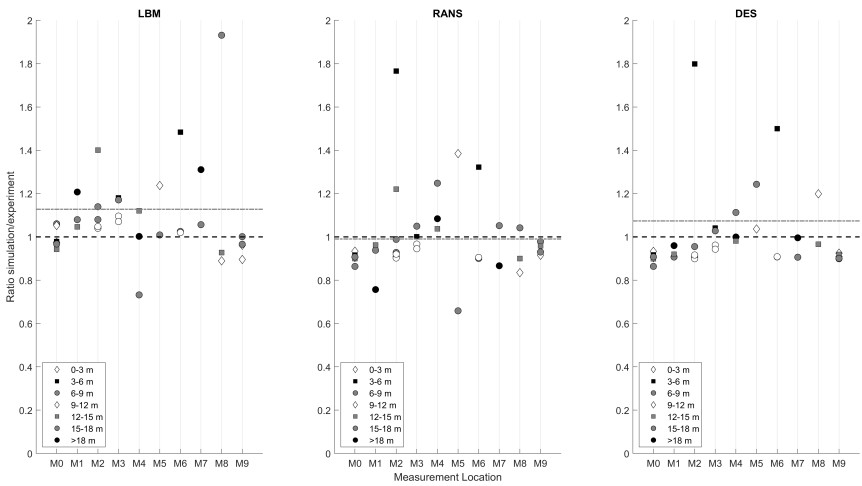

**Figure 8.** Ratio of simulation results to experimental wind speeds as function of measurement location. The dotted grey line represents the average value.

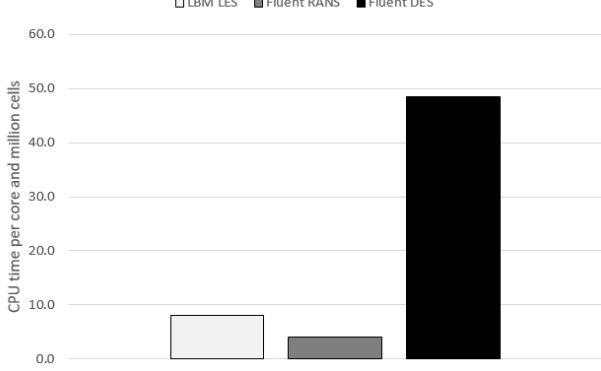

**Figure 9.** Comparison of computational time per cpu core and million cells

RANS simulation. This is due to its explicit formulation and exact advection operator. Furthermore, each of the collision and streaming processes are independent at each lattice, which makes the method so suitable for parallelisation. This advantage extends also to other types of high performance hardware like Graphics Processing Units (GPUs). Some studies of GPUs-based LBM solvers show promising results in this field (Asmuth et al. (2019), Schönherr et al. (2011), Onodera and Idomura (2018)).

5    The performance of this LBM simulation could be increased by adapting the code to use different grid sizes, depending on the flow and therefore reducing the overall cell count drastically. Work on this is ongoing.



**Table 3.** Computational Time. All Simulation were run on 80 cores (Intel Xeon E5-2630V4: 2.2 GHz)

|  | Palabos | Fluent RANS | Fluent DES |
|---|---|---|---|
| Formulation | unsteady | steady | unsteady |
| Cell Count | 41'585'372 | 10'055'540 | 10'055'540 |
| Total CPU Time | 40273.6 | 4821.8 | 58509.7 |
| Seconds/(core · million cells) | 8.1 | 4.0 | 48.5 |

## 5   Conclusion

Accurate modelling of flow features in complex terrain is key for the wind resource assessment. LES has shown potential advantages for modelling the atmospheric boundary layer in previous work; however, is far too computationally expensive to be applied outside the research environment. In this study, a LES simulation using the LBM framework Palabos was implemented to calculate the wind field over the complex terrain of the Bolund Hill. Advantages of LBM include its efficiency, near ideal scalability on High Performance Computers (HPC) and the capabilities to easily automate the geometry, the mesh generation

and the post-processing of the geometry.

The results were compared to RANS and DES simulations using ANSYS Fluent and field measurements. In general there was a good agreement between simulation and experimental data. Some deviations could be observed near the ground, close to the top of cliff (M2) and on the lee side of the hill (M8).

It is perhaps surprising that LBM produces good results despite lacking modeling of surface roughness and the turbulent

boundary layer. This shows that one needs to be careful when applying roughness boundary conditions in N-S, as they can actually make the results less reliable. Furthermore, the intrinsic advantages of LBM are more important than the boundary conditions in this case.

The computational costs of these three models were compared and it has been shown that LBM, even in this not-yet fully optimised set-up of the simulation, can perform 5 times faster than DES and lead to reasonably accurate results.

It can be summarised that LBM has a very high potential for modelling wind flow over complex terrain accurately and at relatively low costs, compared to solving the N-S conventionally. Further studies on other sites are ongoing.

*Author contributions.*   The contribution of the authors in this paper is:

  – Alain Schubiger: carrying out and analysing the simulations.

– Sarah Barber: project management and paper correction.

  – Henrik Nordborg: supervision of Alain Schubiger and paper correction.



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
