# Peer review of "Evaluation of the Lattice Boltzmann Method for wind modelling in complex terrain"

_Wind Energy Science, 2019_

## Referee Comment (RC1) · Anonymous Referee #1 · 19 Feb 2020

Review of WES-2019-106, " Evaluation of the lattice Boltzmann method for wind modelling in complex terrain". by Schubiger et al.

Over all, this manuscript has good quality. The description and presentation of the results are very clear. I have some small comments:

Abstract, Line 5, Please spell out the acronym WAsP .

Abstract, Line 6, LBM is a mesoscopic level method, not microscopic method if one follows the standard definition.

In Introduction section, there are three articles should be cited. One and two are LES for Bolund Hill. The third is using MRT-LBM large eddy simulation for a stable stratified flow over a ridge for a laboratory test case.

[Figure]

Ma, Y., Liu, H. Large-Eddy Simulations of Atmospheric Flows Over Complex Terrain Using the Immersed-Boundary Method in the Weather Research and Forecasting Model. Boundary-Layer Meteorol 165, 421–445 (2017). https://doi.org/10.1007/s10546-017-0283-9.

DeLeon, R., Sandusky, M. & Senocak, I. Simulations of Turbulent Flow Over Complex Terrain Using an Immersed-Boundary Method. Boundary-Layer Meteorol 167, 399–420 (2018). https://doi.org/10.1007/s10546-018-0336-8

Wang, Y., MacCall, B.T., Hocut, C.M. Zeng, X, Fernando, H.J.S. Simulation of stratified flows over a ridge using a lattice Boltzmann model. Environ Fluid Mech (2018). https://doi.org/10.1007/s10652-018-9599-3

Section 2.2, In your regularized-BGK LBM method, do you compute the strain rate tensor for LES using the fluid particle PDF? If so, it is worthwhile to write out the equations for computing the strain rate based on the PDF of the particle because this is critically important.

Section 3.1, In this section, you noted three westerly wind observational cases. It is probably good to point out that only the 270o case was simulated. It is also good to add some description on lateral (North, South) and outflow boundary conditions.

How about the turbulence (such as TKE) comparison? That will substantially improve the paper.

---

## Referee Comment (RC2) · Anonymous Referee #2 · 13 Mar 2020

General comments:

This paper deals with the evaluation of an LBM method for modelling the neutral atmospheric boundary layer over complex terrain. I like the idea to promote LBM for wind energy applications, and found the paper interesting and of overall good quality. Presented results are very encouraging. Although I have some remarks regarding the methodology (see specific comments).

Specific comments:

The main drawback of this paper lies in the differences between the models that are compared. I understand this is an evaluation of the LBM method, and thus it comes with its own limitations (no terrain fitted meshes in this case). However, there are

large differences in the meshes that are used (fully cartesian versus vertically stretched meshes), but also mesh sizes (quite different cell numbers, probably due to the stretching applied in the NS solver? Why not using some mesh "coarsening" with Palabos?), the turbulence models (LES vs DES), boundary conditions (staircase vs terrain fitted). Under these conditions, it is difficult to compare the models and draw conclusions (thinking about the conclusion regarding the use of roughness boundary conditions in NS solvers). Although it can be understood that solvers are intrinsically different and methodologies adapted to each solver have been used, I think it could have been interesting to reduce the differences when possible, comparing the models using the same meshes (no vertical stretching), similar turbulence models (LES vs LES), and same roughness models (slip and no-slip for the NS solvers).

From my point of view, a first, preliminary study regarding velocity and turbulence intensity profiles on a simple flat terrain could have brought insight to the model comparison, rather than directly addressing the complex terrain case. Even on this complex terrain case, a comparison of the velocity and turbulence intensity profiles (as shown in Bechmann et al.) are missing, and could provide more insight in the comparisons.

I also wonder about the potential of LBM to handle terrain roughness. The authors used wall-slip conditions for the ocean and no-slip conditions on land with the LBM solver. Isn't it possible to account for the terrain roughness more precisely, using partial-slip boundary conditions? Is the use of a logarithmic profile at the inlet sufficient to model an ABL? Some insight would be welcome.

One last point that is missing is the choice of the collision model. A discussion is proposed regarding the different possibilities (SRT, MRT, entropic, etc.), but the choice is made to use the standard BGK model, which is not supposed to be the most stable. Moreover, the choice of the relaxation parameter "tau" is not discussed (i.e. is equation 6 fully respected?). A small discussion on the non-dimensioning procedure could also be interesting.

[Figure]

Finally, the "code and data availability" section is not present. Can the Palabos simulation setup be shared with the community? It would probably help researchers to get more familiar with LBM and its application to wind energy.

- Page 1 Line 13: LBM is said to have a particular ability to automate the geometry. The argument is often retained to promote LBM methods. However, I do not see the advantage of LBM in comparison to cartesian-grid Navier-Stokes solver with immersed boundaries. Can the authors comment on this point?

- Page 5 Line 12: the authors should be more specific regarding the value of the Smagorinsky constant they have used. Also, is it the same Smagorinsky model used in the NS solver?

- Page 6 Line 22: more details should be given regarding the inflow turbulence generation. Is the same methodology used in the NS solver?

- Page 7 Line 8: some details regarding the mesh used for NS simulations are given. From my understanding, the mesh is wall-adapted. The authors should make it clear.

- Page 7 Line 20: average results of the DES simulation should also be shown.

- Page 13 Line 9: I think this conclusion should be argued, and, from my point of view, is not receivable. There are too many differences in the models to draw such a conclusion (different meshes, turbulence models?, different wall boundary conditions, etc.)

- Page 13 Line 13: LBM is said to be 5 times faster than DES. However, the total CPU time is only 30% lower. Perhaps a comment on the mesh size reduction that could be obtained(using mesh refinement techniques) would help clarify the potential of LBM methods to reduce CPU time. Anyway, would it be possible to have similar meshes between LBM and NS even using mesh refinment, and, have LBM solvers the same mesh size requierements than NS solvers?

Technical corrections:

Figures text size should be made uniform in the different plots. In the current version, fontsize is obviously too small to be readable (Figs 1, 4, 5, 6, 7, 8).

Page 1 Line 15 : doubled dots

Page 2 Line 22 : a extremely fine –> an extremely fine

Page 4 Eq. 5 : "with" in italic and attached to "f_i"

Page 6 Line 40 : "to an total" –> "to a total"

Page 9 Line 12: a reference to the figure should be added

Page 11 Line 5: space between "et al." and parenthesis.

Page 11 Line 6: "summarise" –> "summarize"

Page 13 Line 2: "is far" –> "it is far" or "LES is far", or replace "; however" with something else to improve readability

Page 13 Line 8: "of cliff" –> "of the cliff"

---

## Author Response (AR1)

Review of WES-2019-106, " Evaluation of the lattice Boltzmann method for wind modelling in complex terrain". by Schubiger et al.

Over all, this manuscript has good quality. The description and presentation of the results are very clear. I have some small comments:

Abstract, Line 5, Please spell out the acronym WAsP .

Abstract, Line 6, LBM is a mesoscopic level method, not microscopic method if one follows the standard definition.

In Introduction section, there are three articles should be cited. One and two are LES for Bolund Hill. The third is using MRT-LBM large eddy simulation for a stable stratified flow over a ridge for a laboratory test case.

Ma, Y., Liu, H. Large-Eddy Simulations of Atmospheric Flows Over Complex Terrain Using the Immersed-Boundary Method in the Weather Research and Forecasting Model. Boundary-Layer Meteorol 165, 421–445 (2017). https://doi.org/10.1007/s10546-017-0283-9.

DeLeon, R., Sandusky, M. & Senocak, I. Simulations of Turbulent Flow Over Complex Terrain Using an Immersed-Boundary Method. Boundary-Layer Meteorol 167, 399–420 (2018). https://doi.org/10.1007/s10546-018-0336-8

Wang, Y., MacCall, B.T., Hocut, C.M. Zeng, X, Fernando, H.J.S. Simulation of stratified flows over a ridge using a lattice Boltzmann model. Environ Fluid Mech (2018). https://doi.org/10.1007/s10652-018-9599-3

Section 2.2, In your regularized-BGK LBM method, do you compute the strain rate tensor for LES using the fluid particle PDF? If so, it is worthwhile to write out the equations for computing the strain rate based on the PDF of the particle because this is critically important.

Section 3.1, In this section, you noted three westerly wind observational cases. It is probably good to point out that only the 270o case was simulated. It is also good to add some description on lateral (North, South) and outflow boundary conditions.

How about the turbulence (such as TKE) comparison? That will substantially improve the paper.

Wind Energ. Sci. Discuss.,
https://doi.org/10.5194/wes-2019-106-RC2, 2020

[Figure]

This paper deals with the evaluation of an LBM method for modelling the neutral atmospheric boundary layer over complex terrain. I like the idea to promote LBM for wind energy applications, and found the paper interesting and of overall good quality. Presented results are very encouraging. Although I have some remarks regarding the methodology (see specific comments).

Specific comments:

The main drawback of this paper lies in the differences between the models that are compared. I understand this is an evaluation of the LBM method, and thus it comes with its own limitations (no terrain fitted meshes in this case). However, there are

large differences in the meshes that are used (fully cartesian versus vertically stretched meshes), but also mesh sizes (quite different cell numbers, probably due to the stretching applied in the NS solver? Why not using some mesh "coarsening" with Palabos?), the turbulence models (LES vs DES), boundary conditions (staircase vs terrain fitted). Under these conditions, it is difficult to compare the models and draw conclusions (thinking about the conclusion regarding the use of roughness boundary conditions in NS solvers). Although it can be understood that solvers are intrinsically different and methodologies adapted to each solver have been used, I think it could have been interesting to reduce the differences when possible, comparing the models using the same meshes (no vertical stretching), similar turbulence models (LES vs LES), and same roughness models (slip and no-slip for the NS solvers).

From my point of view, a first, preliminary study regarding velocity and turbulence intensity profiles on a simple flat terrain could have brought insight to the model comparison, rather than directly addressing the complex terrain case. Even on this complex terrain case, a comparison of the velocity and turbulence intensity profiles (as shown in Bechmann et al.) are missing, and could provide more insight in the comparisons.

I also wonder about the potential of LBM to handle terrain roughness. The authors used wall-slip conditions for the ocean and no-slip conditions on land with the LBM solver. Isn't it possible to account for the terrain roughness more precisely, using partial-slip boundary conditions? Is the use of a logarithmic profile at the inlet sufficient to model an ABL? Some insight would be welcome.

One last point that is missing is the choice of the collision model. A discussion is proposed regarding the different possibilities (SRT, MRT, entropic, etc.), but the choice is made to use the standard BGK model, which is not supposed to be the most stable. Moreover, the choice of the relaxation parameter "tau" is not discussed (i.e. is equation 6 fully respected?). A small discussion on the non-dimensioning procedure could also be interesting.

Finally, the "code and data availability" section is not present. Can the Palabos simulation setup be shared with the community? It would probably help researchers to get more familiar with LBM and its application to wind energy.

- Page 1 Line 13: LBM is said to have a particular ability to automate the geometry. The argument is often retained to promote LBM methods. However, I do not see the advantage of LBM in comparison to cartesian-grid Navier-Stokes solver with immersed boundaries. Can the authors comment on this point?

- Page 5 Line 12: the authors should be more specific regarding the value of the Smagorinsky constant they have used. Also, is it the same Smagorinsky model used in the NS solver?

- Page 6 Line 22: more details should be given regarding the inflow turbulence generation. Is the same methodology used in the NS solver?

- Page 7 Line 8: some details regarding the mesh used for NS simulations are given. From my understanding, the mesh is wall-adapted. The authors should make it clear.

- Page 7 Line 20: average results of the DES simulation should also be shown.

- Page 13 Line 9: I think this conclusion should be argued, and, from my point of view, is not receivable. There are too many differences in the models to draw such a conclusion (different meshes, turbulence models?, different wall boundary conditions, etc.)

- Page 13 Line 13: LBM is said to be 5 times faster than DES. However, the total CPU time is only 30% lower. Perhaps a comment on the mesh size reduction that could be obtained(using mesh refinement techniques) would help clarify the potential of LBM methods to reduce CPU time. Anyway, would it be possible to have similar meshes between LBM and NS even using mesh refinment, and, have LBM solvers the same mesh size requierements than NS solvers?

Technical corrections:

Figures text size should be made uniform in the different plots. In the current version, fontsize is obviously too small to be readable (Figs 1, 4, 5, 6, 7, 8).

Page 1 Line 15 : doubled dots

Page 2 Line 22 : a extremely fine –> an extremely fine

Page 4 Eq. 5 : "with" in italic and attached to "f_i"

Page 6 Line 40 : "to an total" –> "to a total"

Page 9 Line 12: a reference to the figure should be added

Page 11 Line 5: space between "et al." and parenthesis.

Page 11 Line 6: "summarise" –> "summarize"

Page 13 Line 2: "is far" –> "it is far" or "LES is far", or replace "; however" with something else to improve readability

Page 13 Line 8: "of cliff" –> "of the cliff"

**Point-by-point response**

**RC1**

- Abstract, Line 5, Please spell out the acronym WAsP:

changed in the new version of the paper

- Abstract, Line 6, LBM is a mesoscopic level method, not microscopic method if one follows the standard definition.

changed in the new version of the paper

- In Introduction section, there are three articles should be cited.

added to the new version of the paper

- Section 2.2, In your regularized-BGK LBM method, do you compute the strain rate tensor for LES using the fluid particle PDF?

The implemented regularization process is best described in this work of Jonas Latt and Bastien Chopard. Latt, Jonas, and Bastien Chopard. "Lattice Boltzmann method with regularized pre-collision distribution functions." Mathematics and Computers in Simulation 72.2-6 (2006): 165-168. A reference was added to the new version of the paper.

- Section 3.1, ... It is also good to add some description on lateral (North, South) and outflow boundary conditions.

Description was added to the new version

- How about the turbulence (such as TKE) comparison?

A comparion of the turbulence has been added to the new version of the paper

**RC2**

- "quite different cell numbers…":

Exactly, the Fluent mesh was created with the Fluent meshing tool and therefore there were more possibilities to implement local adjustments. It has been added to the paper.

- "Why not use mesh coarsening with Palabos":

Studies that apply the grid refinement capabilities of Palabos were not within the scope of this first study; however, this is will be tested in the future. It has been added to the set-up description.

- "From my point of view……":

As previous studies have shown that Palabos works well for turbulent flows (Wissocq, Gauthier, et al. "Regularized characteristic boundary conditions for the Lattice-Boltzmann methods at high Reynolds number flows." Journal of Computational Physics 331 (2017): 1-18.), the aim of this study was to specifically test the applicability to wind energy. Bolund Hill was chosen for this due to the quality of available measurement data. It is correct to say that a simpler geometry may have been easier to start with, and we are considering further comparison cases.

- "….turbulence intensity profiles….":

A comparison of the turbulence has been added to the new version of the paper

- "Isn't it possible to account for terrain roughness":

Unfortunately it is not easily possible to account for different surface roughnesses in Palabos at this point. This mentioned on line 11 on page 6. This is a topic that we are planning to investigate in the future.

- "Is the use of a logarithmic profile sufficient?":

We were following the guidelines of the Bolund Hill Blind Test and used the provided logarithmic velocity profile, which were fitted to the measurements.

- "…collision model… The choice of Tau ":

The BGK model was chosen for simplicity in this first study. Tau respectively Nu were chosen so we could achieve a stable solution and eq. 6 is respected. This has been described in the new version of the paper

- "A small discussion on the non-dimensioning procedure…":

A reference has been added to the new version.
Latt, Jonas. "Choice of units in lattice Boltzmann simulations." Freely available online at http://lbmethod.org/_media/howtos: lbunits.pdf (2008).

- "Can the Palabos simulation setup be shared with the community":

The code is available on the git repository. The link has been added to the new version of the paper.

- "Page 1 Line 13: LBM is said to have a particular ability…":

It is true. With regard to mesh and geometry generation, the difference to cartesian-grid NS solver with immersed boundaries is not that great, as compared to wall-adapted NS solvers. However, the main advantages LBM like intrinsic massive parallelism or offering simplicity in development are present

- "Page 5, Line 12: Smagorinsky model":

The Smagorinksy constant was set to 0.14. This has been added to the new version of the paper

- "Page 6, Line 22: Is the same methology used in the NS solver?":

The Fluent setup uses the Synthetic Turbulence Generator scheme. For the LBM simulation we implemented a rudimentary transient inflow condition. The Fluent implementation is much more sophisticated. This is described on page 6 and 7 of the paper".

- "Page 7, Line 20: average results of the DES simulation should also be shown.":

This has been added to the new version of the paper

- "Page 13, Line 9: I think this conclusion should be argued, and, from my point of view, is not receivable.":

Parts of the conclusion have been adjusted and the difference in the simulation approaches have been made clearer. For example the summary: It can be summarised that LBM may be applicable to modelling wind flow over complex terrain accurately at relatively low costs if the challenges raised in this work are addressed. Further studies on other sites are ongoing.

- "Page 13, Line 13: "Perhaps a comment on the mesh size reduction…":

An estimation on the mesh size reduction has been added.

- "would it be possible to have similar meshes between LBM and NS even using mesh refinment, and, have LBM solvers the same mesh size requierements than NS solvers?":

More or less this is possible and is part of a running follow-up project.

- Figure text size and following comments: changed

Page 1 Line 15 : doubled dots

Page 2 Line 22 : a extremely fine –> an extremely fine

Page 4 Eq. 5 : "with" in italic and attached to "f_i"

Page 6 Line 40 : "to an total" –> "to a total"

Page 9 Line 12: a reference to the figure should be added

Page 11 Line 5: space between "et al." and parenthesis.

Page 11 Line 6: "summarise" –> "summarize"

Page 13 Line 2: "is far" –> "it is far" or "LES is far", or replace "; however" with something

else to improve readability

Page 13 Line 8: "of cliff" –> "of the cliff"

Wind Energ. Sci. Discuss.,
https://doi.org/10.5194/wes-2019-106-AC1, 2020

[Figure]

Abstract, Line 5, Please spell out the acronym WAsP: changed in the new version of the paper

Abstract, Line 6, LBM is a mesoscopic level method, not microscopic method if one follows the standard definition. changed in the new version of the paper

In Introduction section, there are three articles should be cited. added to the new version of the paper

Section 2.2, In your regularized-BGK LBM method, do you compute the strain rate

tensor for LES using the fluid particle PDF?

The implemented regularization process is best described in this work of Jonas Latt and Bastien Chopard. Latt, Jonas, and Bastien Chopard. "Lattice Boltzmann method with regularized pre-collision distribution functions." Mathematics and Computers in Simulation 72.2-6 (2006): 165-168. A reference was added to the new version of the paper.

Section 3.1, ... It is also good to add some description on lateral (North, South) and outflow boundary conditions. Description was added to the new version

How about the turbulence (such as TKE) comparison? A comparion of the turbulence has been added to the new version of the paper

Wind Energ. Sci. Discuss.,
https://doi.org/10.5194/wes-2019-106-AC2, 2020

[Figure]

Exactly, the Fluent mesh was created with the Fluent meshing tool and therefore there were more possibilities to implement local adjustments. It has been added to the paper.

- "Why not use mesh coarsening with Palabos":

Studies that apply the grid refinement capabilities of Palabos were not within the scope of this first study; however, this is will be tested in the future. It has been added to the set-up description.

- "From my point of view. . . . . .":

As previous studies have shown that Palabos works well for turbulent flows (Wissocq, Gauthier, et al. "Regularized characteristic boundary conditions for the Lattice-Boltzmann methods at high Reynolds number flows." Journal of Computational Physics 331 (2017): 1-18.), the aim of this study was to specifically test the applicability to wind energy. Bolund Hill was chosen for this due to the quality of available measurement data. It is correct to say that a simpler geometry may have been easier to start with, and we are considering further comparison cases.

- ". . ..turbulence intensity profiles. . ..":

A comparison of the turbulence has been added to the new version of the paper

- "Isn't it possible to account for terrain roughness":

Unfortunately it is not easily possible to account for different surface roughnesses in Palabos at this point. This mentioned on line 11 on page 6. This is a topic that we are planning to investigate in the future.

- "Is the use of a logarithmic profile sufficient?":

We were following the guidelines of the Bolund Hill Blind Test and used the provided logarithmic velocity profile, which were fitted to the measurements.

- ". . .collision model. . . The choice of Tau ":

The BGK model was chosen for simplicity in this first study. Tau respectively Nu were chosen so we could achieve a stable solution and eq. 6 is respected. This has been described in the new version of the paper

- "A small discussion on the non-dimensioning procedure. . .":

A reference has been added to the new version. Latt, Jonas. "Choice of units in lattice Boltzmann simulations." Freely available online at http://lbmethod.org/_media/howtos:

lbunits.pdf (2008).

- "Can the Palabos simulation setup be shared with the community":

The code is available on the git repository. The link has been added to the new version of the paper.

- "Page 1 Line 13: LBM is said to have a particular ability...":

It is true. With regard to mesh and geometry generation, the difference to cartesian-grid NS solver with immersed boundaries is not that great, as compared to wall-adapted NS solvers. However, the main advantages LBM like intrinsic massive parallelism or offering simplicity in development are present

- "Page 5, Line 12: Smagorinsky model":

The Smagorinksy constant was set to 0.14. This has been added to the new version of the paper

- "Page 6, Line 22: Is the same methology used in the NS solver?":

The Fluent setup uses the Synthetic Turbulence Generator scheme. For the LBM simulation we implemented a rudimentary transient inflow condition. The Fluent implementation is much more sophisticated. This is described on page 6 and 7 of the paper".

- "Page 7, Line 20: average results of the DES simulation should also be shown.":

This has been added to the new version of the paper

- "Page 13, Line 9: I think this conclusion should be argued, and, from my point of view, is not receivable.":

Parts of the conclusion have been adjusted and the difference in the simulation approaches have been made clearer. For example the summary: It can be summarised that LBM may be applicable to modelling wind flow over complex terrain accurately at relatively low costs if the challenges raised in this work are addressed. Further studies

on other sites are ongoing.

- "Page 13, Line 13: "Perhaps a comment on the mesh size reduction...":

An estimation on the mesh size reduction has been added.

- "would it be possible to have similar meshes between LBM and NS even using mesh refinment, and, have LBM solvers the same mesh size requierements than NS solvers?":

More or less this is possible and is part of a running follow-up project.

Figure text size and following comments: changed

Page 1 Line 15 : doubled dots

Page 2 Line 22 : a extremely fine –> an extremely fine

Page 4 Eq. 5 : "with" in italic and attached to "f_i"

Page 6 Line 40 : "to an total" –> "to a total"

Page 9 Line 12: a reference to the figure should be added

Page 11 Line 5: space between "et al." and parenthesis.

Page 11 Line 6: "summarise" –> "summarize"

Page 13 Line 2: "is far" –> "it is far" or "LES is far", or replace "; however" with something else to improve readability

Page 13 Line 8: "of cliff" –> "of the cliff"

[revised manuscript text omitted]
. ~~In this paper, Fluent was first set up to solve the Reynolds-Averaged Navier-Stokes (RANS) equations. The RANS equations govern the transport of the averaged flow quantities, with the whole range of the scales of turbulence being modeled. The RANS based modeling approach therefore greatly reduces the required computational effort and resources, and is widely adopted for practical engineering applications. The k-epsilon turbulence model was applied to attain closure. This differs from the Large Eddy Simulation (LES) approach, which explicitly computes large eddies in a time-dependent simulation using the filtered' N-S equations. The rationale behind LES is that by modeling less of turbulence (and resolving more), the error introduced by turbulence modeling can be reduced. It is also believed to be easier to find a "universal" model for the small scales, since they tend to be more isotropic and less affected by the macroscopic features like boundary conditions, than the large eddies. Filtering is essentially a mathematical manipulation of the exact N-S equations to remove the eddies that are smaller than the size of the filter, which is usually taken as the mesh size. Like Reynolds-averaging, the filtering process creates additional unknown terms that must be modeled to achieve closure. Statistics of the time-varying flow-fields such as time-averages and r.m.s. values of the solution variables, which are generally of most engineering interest, can be collected during the time-dependent simulation.~~

[revised manuscript text omitted]